# Consequences of Domestication on Gut Microbiome: A Comparative Analysis Between Wild Boars and Domestic Pigs

**DOI:** 10.3390/ani15050747

**Published:** 2025-03-05

**Authors:** Da-Yun Bae, Sung-Hyun Moon, Taek Geun Lee, Young-Seung Ko, Yun-Chae Cho, Hamin Kang, Chan-Soo Park, Jung-Sun Kang, Yeonsu Oh, Ho-Seong Cho

**Affiliations:** 1College of Veterinary Medicine and Bio-Safety Research Institute, Jeonbuk National University, Iksan 54596, Republic of Korea; bdy7700@naver.com (D.-Y.B.); chunsu17@naver.com (S.-H.M.); 43674@jbnu.ac.kr (T.G.L.); dudtmd3315@naver.com (Y.-S.K.); cyc437@gmail.com (Y.-C.C.); 2Genebiotech Co., Ltd., Gongju 32619, Republic of Korea; hmkang@genebiotech.co.kr (H.K.); cspark@genebiotech.co.kr (C.-S.P.); jskang@genebiotech.co.kr (J.-S.K.); 3College of Veterinary Medicine and Institute of Veterinary Science, Kangwon National University, Chuncheon 24341, Republic of Korea

**Keywords:** domestic pig, gut microbiome, next-generation sequencing, probiotic genera, wild boar

## Abstract

Gut microbiota plays a vital role in animal health, impacting digestion, immunity, and overall well-being. Domestic pigs (*Sus scrofa domesticus*) and wild boars (*Sus scrofa scrofa*) share a common ancestor, yet their distinct environments and diets have likely influenced their gut microbial composition. This study compares the gut microbiota of wild boars and domestic pigs using the next-generation sequencing of 16S rRDA genes. The results show that beneficial bacteria such as *Bifidobacterium* and *Lactobacillus*, are more abundant in wild boars. Functional analyses suggest that wild boars possess microbiota associated with enhanced immune responses and environmental adaptability. In our research, we investigated the distinctions in intestinal microbiota between wild boars and domestic pigs to identify microorganisms absent in the latter. Our study uncovered lactobacilli strains unique to wild boars, which present potential for development as probiotics.

## 1. Introduction

The mammalian gastrointestinal tract harbors a complex ecosystem comprising approximately 1000 identified bacterial species [1]. These microbial communities, collectively termed the gut microbiota, play crucial roles in host physiology, influencing nutrition, immunity, and even mental health through intricate interactions with the host and other microorganisms [1,2]. This symbiotic relationship provides mutual benefits: the host offers a nutrient-rich, stable environment, while the microbiota assists in extracting energy from otherwise indigestible food components and synthesizing essential metabolites [3].

Traditionally, studies on gut microbiota relied heavily on culture-dependent methods. However, these approaches faced limitations due to the inherent unculturability of nearly half of the gut microbial species in vitro [4]. The advent of high-throughput sequencing (HTS), particularly 16S rRNA gene-based techniques, revolutionized the field by enabling culture-independent analyses. These methods provide a detailed characterization of gut microbial communities, including previously unculturable species, significantly expanding our understanding of gut microbiota dynamics and ecological interactions [5,6].

Research on swine gut microbiota has steadily progressed, revealing that microbial composition is shaped by various factors, including breed and diet [7,8]. While modern pigs and wild boars share a common ancestor, domestication has led to significant divergences in their lifestyles, diets, and physiological characteristics [9]. Wild boars, thriving in natural habitats, consume fibrous diets rich in cellulose, such as acorns, wild fruits, and roots. This natural diet is hypothesized to contribute to their robust environmental adaptability and disease resistance [10,11]. Conversely, it is suggested that domestic pigs, which are typically fed simplified, high-carbohydrate diets, may experience reduced disease resistance, elongated intestines, and accelerated growth rates [12].

A prevailing hypothesis suggests that domestication, particularly the shift to controlled diets, may have resulted in the loss of beneficial gut bacteria such as *Bifidobacterium* and *Lactobacillus* in domestic pigs. *Bifidobacterium* aids in maintaining the intestinal barrier, helps digest food, and stimulates the immune system, while *Lactobacillus* assists in the digestion of sugars, inhibits harmful bacteria through the production of lactic acid, and is crucial for synthesizing B vitamins and enhancing mineral absorption [13]. Despite the recognized importance of this topic, comparative studies investigating the gut microbiota of wild boars and domestic pigs remain limited. This study addresses this knowledge gap by conducting a comparative analysis of their gut microbiota using NGS of 16S r RNA genes. The primary objective is to identify potentially beneficial microbes that may have been lost during the domestication process, offering insights into the evolutionary and functional consequences of dietary changes on gut microbial communities.

## 2. Materials and Methods

### 2.1. Animals and Sample Collection

A total of 69 fecal samples were collected from animals categorized into two groups: 34 wild boars (WB) and 35 domestic pigs (DP) (Table 1). In South Korea, domestic pigs are typically fed a diet consisting of grains like corn and barley, soybean meal for protein, and added vitamins and minerals to support healthy growth and immune function. Fecal samples from wild boars were obtained from captive individuals in three provinces: South Korea–Jeonnam Province, Gyeongnam Province, and Gangwon Province. In contrast, domestic pig fecal samples were sourced from two commercial breeding farms. The domestic pigs were selected to represent various growth stages, including weanling, grower, gestating sows, and lactating sows.

To ensure sample integrity, fecal matter was collected from the uppermost layer, which had not come into contact with the ground, minimizing contamination by soil bacteria. All samples were handled using sterilized equipment, placed into sterile containers, and immediately stored at −20 °C. They were then transported under controlled temperature to the laboratory, where they underwent DNA extraction and sequencing.

### 2.2. Total Bacterial Genomic DNA (gDNA) Extraction

Genomic DNA (gDNA) was extracted using the QIAamp PowerFecal Pro DNA Kit (QIAGEN, Hilden, Germany) following the manufacturer’s protocol. In brief, 250 mg of stool was homogenized in 800 µL of lysis buffer with a bead-beating homogenization step. The mixture was then centrifuged at 15,000× *g* for 1 min, and the resulting supernatant was processed through binding and washing steps. DNA was eluted using 50 µL of elution buffer.

The quality and quantity of the extracted DNA were assessed using an Epoch microplate spectrophotometer (BioTek Instruments, Winooski, VT, USA). The samples were subsequently stored at −20 °C until they were ready for NGS.

### 2.3. PCR Amplification and Sequencing

The V4 hypervariable region of the bacterial 16S rRNA gene was amplified using the primers 515F (5′-GTGCCAGCMGCCGCGGTAA-3′) and 806R (5′-GGACTACHVGGGTWTCTAAT-3′) [14]. The amplification reaction was conducted in a 25 µL volume containing 2.5 µL of 10× Ex Taq buffer, 2 µL of 2.5 mM dNTP mix, 0.25 µL of Takara Ex Taq DNA Polymerase (5 U/µL), 2 µL of each primer (10 pM), and 4 µL of extracted gDNA. Nuclease-free water was added to reach the total volume. The PCR conditions included an initial denaturation step at 95 °C for 3 min, followed by 25 cycles of denaturation at 95 °C for 30 s, annealing at 55 °C for 30 s, and an extension at 72 °C for 30 s, with a final elongation step at 72 °C for 5 min.

PCR products were purified using the magnetic bead-based Agencourt AMPure XP Reagent (Beckman Coulter Inc., Brea, CA, USA) and verified by electrophoresis on a 1% agarose gel. Samples displaying a bright primary band (approximately 400–450 bp) were selected for subsequent experiments. The PCR reaction incorporated fusion primers containing sequences for Illumina adapters, multiplexing barcodes, and the target-specific regions as outlined on the provided Chunlab Bacterial V4 Fusion primer list (https://help.ezbiocloud.net/wp-content/uploads/2019/02/Chunlab_Bacterial-V4-Fusion-primer-list-3.xlsx) (accessed on 15 March 2024).

Purified libraries were quantified using a Qubit 2.0 fluorometer (Thermo Fisher Scientific Inc., Waltham, MA, USA). Libraries were then pooled in equimolar concentrations and sequenced on an iSeq 100 Sequencing system (Illumina, San Diego, CA, USA) using a single-end (1 × 300 bp) sequencing protocol. The phiX control was added to the sequencing run at a concentration of 10%, and the sample concentration loaded into the iSeq100 cartridge was 50 pM to ensure optimal cluster density and data quality [15].

### 2.4. Bioinformatics Analysis

A microbiome analysis of the V4 hypervariable region of the 16S rRNA gene was conducted using the Quantitative Insights Into Microbial Ecology 2 (QIIME2) software (Version 2023.2) [16]. Following Illumina sequencing, demultiplexed reads were processed using the DADA2 plugin (version 1.26.0) [17], which removed ambiguous, redundant, low-quality, and chimera sequences. Taxonomic classification of the quality-filtered sequence was performed with the SILVA database (Version 138) and the sklearn classifier [18]. Bacterial compositions were categorized at taxonomic levels ranging from kingdom to species. Comparative analyses between DP and WB samples focused on the phylum, family, and genus levels. Statistical analysis and data mining were carried out using the microeco R package and functions (Version 4.2.0) [19].

To identify unique and shared operational taxonomic units (OTUs) across groups, Venn diagrams were constructed to compare the population distribution at various taxonomic levels. Alpha diversity indices, namely Shannon–Wiener, Chao1, and Observed Species, were calculated and presented in box-and-whisker plots. Beta diversity was assessed using Principal Coordinate Analysis (PCoA) based on Bray–Curtis dissimilarity to illustrate the differences in bacterial community composition between the two groups.

Taxonomic composition and relative abundance were analyzed at the phylum, family, and genus levels and presented as stacked bar plots. Differentially abundant taxa were identified using the DESeq2 method and linear discriminant analysis (LDA) effect size (LEfSe) with LDA values > 4 [20,21].

The functional potential of the gut microbiota was inferred using the Phylogenetic Investigation of Communities by Reconstruction of Unobserved States 2 (PICRUSt2; Version 2.5.2; https://github.com/picrust/picrust2) (accessed on 10 March 2024). Functional pathway differences between the two groups were identified using the Kyoto Encyclopedia of Genes and Genomes (KEGG; http://www.genome.jp/kegg/) (accessed on 10 March 2024). at the second pathway level [22].

## 3. Results

### 3.1. Statistics for Sequencing Data

A total of 69 fecal samples were analyzed, comprising 35 from domestic pigs (DP) and 34 from wild boars (WB). The sequencing of these samples produced 5,568,208 raw reads, with a mean of 80,698.67 reads per sample and a standard deviation of 20,632.86. Following quality filtering, 4,685,995 high-quality reads were retained, resulting in an average of 67,912.97 reads per sample, with a standard deviation of 17,025.67. These high-quality reads were then categorized into 6452 amplicon sequence variants (ASVs).

The Venn diagram (Figure 1) illustrates the distribution of shared and unique OTUs between the two groups. In this analysis, OTUs are employed as a preliminary classification unit within the AmpliSeq approach, functioning as a basis for further taxonomic assignment to species or genus levels. Among the observed OTUs, 3896 were identified in the DP group and 3456 in the WB group. A total of 900 OTUs (48.1%) were shared between the two groups, while 2996 OTUs (31.7%) were unique to the DP group and 2556 OTUs (20.2%) were unique to the WB group.

### 3.2. Bacterial Diversity

Figure 2 illustrates the differences in bacterial diversity between the DP and WB groups, assessed using alpha diversity indices (Observed, Shannon, and Chao 1). Significant differences were observed between the groups: the Observed species index was 551.05 ± 19.20 for DP and 368.35 ± 25.71 for WB, the Shannon index was 4.81 ± 0.08 for DP and 4.14 ± 0.16 for WB, and the Chao 1 index was 584.42 ± 20.60 for DP and 386.87 ± 27.09 for WB (*p* < 0.01 or 0.001).

Beta diversity analysis was conducted to examine differences in bacterial community composition between the groups. The results indicated distinct microbial community structures for DP and WB. This distinction was further visualized in the PCoA plot, which showed clear clustering of each group at distant positions from one another (Figure 3).

### 3.3. Taxonomic Profiles

The taxonomic distributions of the most abundant bacterial OTUs were analyzed at the phylum, family, and genus levels. For each taxonomic rank, the relative abundance of bacteria taxa was presented for individual samples as well as the group mean (Figure 4).

At the phylum level, *Bacillota* were the most predominant in both groups, accounting for 61.66% in WB and 52.62% in DP, followed by *Bacteroidota*, which constituted 19.55% in WB and 36.17% in DP. The combined abundance of *Bacilllota*, *Bacteroidota*, *Pseudomonadota,* and *Actinobacteriota* represented 97.54% of the total microbial composition in WB and 95.17% in DP. Other phyla, including *Spirochaetota*, *Verrucomicrobiota*, *Euryarchaeota*, and *Desulfobacterota*, were less abundant.

At the family level, the most abundant families in WB were *Lachnospiraceae* (18.05%), *Prevotellaceae* (9.81%), *Oscillospiraceae* (7.71%), *Bifidobacteriaceae* (5.58%), and *Enterobacteriaceae* (5.27%). In DP, the dominant families included *Prevotellaceae* (17.31%), *Oscillospiraceae* (14.29%), *Muribaculaceae* (8.71%) *Lachnospiraceae* (7.25%), and *Christensenellaceae* (6.75%).

At the genus level, there was no overlap among the top five genera between the groups. The predominant genera in WB were *Bifidobacterium* (5.53%), *Prevotellaceae_NK3B31_group* (5.51%), *Escherichia-Shigella* (5.2%), *UCG-005* (4.93%), and *Lactobacillus* (4.04%). In DP, the dominant genera were *Muribaculaceae* (8.39%), *Prevotella* (7.95%), *Christensenellaceae_R-7_group* (6.67%), *Prevotellaceae_NK3B31_group* (5.14%), and *NK4A214_group* (4.58%).

DESeq analysis further revealed significant differences in genus abundance between the groups. In DP, the abundance of *Muribaculaceae*, *Prevotella*, *Christensenellaceae_R-7_group*, *NK4A214_group*, and *[Eubacterium]_coprostanoligenes_group* was significantly higher (*p* < 0.01 or 0.001). In contrast, WB exhibited significantly higher levels of *Bifidobacterium*, *Blautia*, *Escherichia-Shigella*, *UCG-008*, and *Lactobacillus* (*p* < 0.001) (Figure 5).

### 3.4. Linear Discriminant Analysis Effect Size (LEfSe) Analysis

LEfSe analysis (LDA score = 4.0, *p* < 0.05) was performed to identify metagenomic biomarkers differentiating the DP and WB groups (Figure 6).

The LEfSe plot (Figure 6a) highlights microbial taxa with significantly different abundances between the two groups. The DP group showed higher relative abundances of 16 taxa, including *Bacteroidota* phylum, *Bacteroidia* class, *Bacteroidales* and *Oscillospirales* orders, and families such as *Prevotellaceae*, *Oscillospiraceae*, *Muribaculaceae*, *Rikenellaceae*, *[Eubacterium]_coprostanoligenes_group*, and *Micrococcaceae.* Genera with increased abundance in DP included *Prevotella*, *Muribaculaceae*, *NK4A214_group*, *UCG-002*, *Rikenellaceae_RC9_gut_group*, and *[Eubacterium]_coprostanoligenes_group.*

In contrast, the WB group exhibited higher relative abundances of 13 taxa, including *Actinobacteriota* phylum, *Micrococcales*, *Bifidobacteriales*, *Lactobacillales,* and *Lachnospirales* orders, as well as *Bacilli* class. Families such as *Butyricicoccaceae*, *Bifidobacteriaceae,* and *Lachnospirceae* were enriched in WB, along with genera like *UCG-008*, *Blautia,* and *Bifidobacterium*.

The cladogram (Figure 6b) further illustrates the most significant biomarkers in DP and WB. Taxa significantly enriched in DP predominantly belonged to the *Bacteroidota* phylum, while those enriched in WB were primarily associated with *Bacillota* and *Actinobacteriota*.

### 3.5. Predicted Functions of Microbiota

PICRUSt analysis was conducted to explore the functional potential of gut bacteria and to characterize the roles of microbiota in the DP and WB groups. A total of 34 KEGG pathways exhibited significant differences between the groups (*p* < 0.05) (Figure 7).

The WB group displayed a higher abundance of microbial genes associated with carbohydrate metabolism, lipid metabolism, xenobiotics biodegradation and metabolism, transport and catabolism, digestive system functions, immune system regulation, membrane transport, and signal transduction.

In contrast, the DP group showed greater proportions of microbial genes linked to amino acid metabolism, nucleotide metabolism, metabolism of cofactors and vitamins, endocrine system functions, replication and repair, transcription, and translation.

## 4. Discussion

This study analyzed the gut microbiota of wild boars and domestic pigs to identify beneficial microbes that may have been lost during domestication and to explore their potential role in improving pig health. Using NGS of 16S rRNA genes, fecal samples from 34 wild boars and 35 domestic pigs were compared for bacterial diversity, taxonomic composition, and functional potential.

Surprisingly, domestic pigs exhibited significantly higher alpha-diversity indices compared to wild boars (*p* < 0.01 or 0.001). This increased microbial diversity is likely due to the nutrient-balanced diets and farming practices in the Korean swine industry. Previous studies have described that rearing conditions and diet are key drivers of gut microbiota composition [23,24,25]. Feed additives, such as lactic acid bacteria, *Bacillus*, and yeast [26,27,28], which are commonly used in pig diets, could also contribute to this enhanced diversity [29]. These findings align with previous studies showing higher alpha diversity in domestic pigs, though wild boars’ lower diversity may reflect their reliance on natural diets [25,30].

Beta diversity analysis revealed distinct microbial community structures between the two groups, with wild boars forming more homogeneous clusters and domestic pigs displaying greater heterogeneity. These differences are consistent with prior studies reporting strong distinctions in bacterial community composition between wild and domestic pigs. Variations in rearing environments and diets likely play a pivotal role in shaping these microbiota profiles [30,31,32]. For example, Wei et al. (2022) noted that wild boars had higher levels of beneficial bacteria such as *Bifidobacterium* and various *Lactobacillus* species, suggesting adaptations to a varied, natural diet [31]. Conversely, domestic pigs showed enhancements in genes related to carbohydrate metabolism, reflecting their simplified, high-energy feed regimes. Additionally, Rahman et al. (2024) observed that while domestication has not led to a loss of microbial species, it has resulted in increased alpha diversity and a surge in opportunistic genera, likely due to intensified hygiene and antibiotic usage in modern swine farming [30]. Similarly, Huang et al. (2020) found distinct microbial compositions linked to diet in wild boars compared to commercial and domestic pigs, underscoring the influence of diet on gut microbiota [32]. These findings collectively illustrate how domestication and the associated changes in diet and environment have profoundly modeled the gut microbiota, differentiating domestic pigs from their wild counterparts.

At the phylum level, *Bacteroidota* was significantly more abundant in domestic pigs, whereas *Actinobacteriota* predominated in wild boars. Genera such as *Prevotella* were key biomarkers in domestic pigs, while *Bifidobacterium* and *Blautia* were more abundant in wild boars. Interestingly, wild boars exhibited a higher prevalence of lactic acid-producing bacteria, including *Lactobacillus*, *Enterococcus*, *Streptococcus*, and *Bifidobacterium*, alongside *Blautia.* These probiotic bacteria play critical roles in enhancing gut health, suppressing pathogens, and supporting immune functions. The higher abundance of these beneficial microbes in wild boars may explain their superior disease resistance [33,34]. *Lactobacillus* contributed to the growth and reduction in diarrhea and the inhibition of pathogens in pigs [35]. *Enterococcus* improved intestinal health in piglets during critical phases such as the weaning period [36]. *Streptococcus*, as one of the most commonly used genera for probiotics, increased growth and feed efficiency in pigs of the weanling stage [37]. *Bifidobacterium* also helps to enhance gut health, immunity, and growth performance in weaning piglets [38]. *Blautia* has, as a genus of the *Lachnospiraceae* family, probiotic characteristics that occur widely in the feces and intestines of mammals. It contributes to alleviating inflammatory diseases and metabolic diseases and to its antibacterial activity against specific microorganisms [39,40].

In contrast, domestic pigs’ gut microbiota exhibited reduced diversity in probiotic bacteria, likely due to simplified diets and breeding including antibiotic use. This imbalance may increase the risk of pathogenic colonization and disease [41]. However, reports of higher *Lactobacillus* abundance in domestic pigs compared to wild boars [13,31,32] suggest that geographical and environmental factors may influence these findings [42,43].

Moreover, the increased prevalence of *Escherichia-Shigella* in wild boars underscores a unique risk of exposure to pathogenic microorganisms in the wild environment [44]. Although *Escherichia-Shigella* species can promote the digestion and absorption of food in wild animals [45], they also contribute to the microbial diversity necessary for dietary fiber breakdown, as observed in studies involving sows where increased fiber intake correlated with a higher relative abundance of these bacteria [46]. Moreover, while *Escherichia-Shigella* is typically associated with microbial ecological imbalance and gastrointestinal disorders, its prevalence can vary seasonally. This is highlighted by Williams et al. (2023), who document seasonal fluctuations—particularly an increase during summer—in *Escherichia-Shigella* populations [44]. Chen et al. (2024) further support this by noting that while there is variability in the diversity of Escherichia-Shigella, other factors such as seasonal variations tend to have a more pronounced impact on these populations than previously considered [47].

Given that our sampling was confined to the spring season, the observed high relative abundance of *Escherichia-Shigella* may reflect seasonal influences and the inherent pathogenic exposure in the wild environment. This environment, unlike that of domestic pigs which are exposed to fewer pathogens, supports a diverse diet and thus a broader spectrum of microbial interactions. The significant presence of *Escherichia-Shigella* in the gut microbiota of wild boars highlights the adaptive responses to their dietary and environmental conditions, underlining the complex interaction between diet, microbial ecology, and pathogen exposure in wild populations. Further investigation is needed to resolve these discrepancies and understand the contextual drivers of microbial variation.

PICRUSt analysis of the microbiota revealed distinct functional profiles between wild boars and domestic pigs. Wild boars’ microbiota enriched genes associated with carbohydrate and lipid metabolism, xenobiotics biodegradation, and various biological processes crucial for adaptation to harsh environments and disease resistance. Conversely, domestic pigs displayed enhanced gene representation linked to amino acid and nucleotide metabolism, as well as functions supporting growth-oriented diets and intensive farming practices, such as cofactors and vitamin synthesis, endocrine regulation, and cellular replication mechanisms [48].

Despite these insights, this study has limitations. The domestic pig samples were collected across various growth stages, whereas wild boars represented a broader age range. Additionally, domestic pig samples were sourced from only two farms in Korea, which may limit the generalizability of the findings. Future studies should address these limitations by incorporating larger, more diverse populations and standardizing sample collection across age groups.

## 5. Conclusions

Our comparative analysis of the gut microbiota in wild boars and domestic pigs through 16S rRNA gene sequencing highlights significant differences driven by domestication. Wild boars exhibit a richer presence of beneficial bacteria such as *Bifidobacterium* and *Lactobacillus*, associated with enhanced immune responses and adaptability, compared to domestic pigs. These differences are likely influenced by natural diets in wild boars and high carbohydrate, antibiotic-influenced diets in domestic pigs.

Functional gene analysis suggests that wild boar microbiota supports robust health through enriched metabolic and immune pathways, while domestic pigs show adaptations that may compromise their disease resistance. This study underscores the potential for using wild boar microbiota as probiotics to improve the health of domestic pigs, suggesting a valuable direction for future agricultural practices.

In summary, this study highlights significant differences in the gut microbiota composition between domestic pigs and wild boars, underscoring the impact of domestication on microbial diversity. Further research is necessary to explore the potential of transferring beneficial microbes from wild to domestic populations.

## Figures and Tables

**Figure 1 animals-15-00747-f001:**
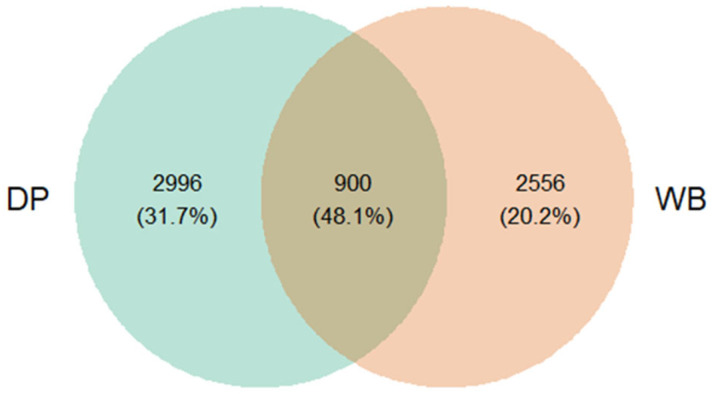
Venn diagram illustrating shared and unique OTUs between DP and WB groups. DP, domestic pig; WB, wild boar. This diagram displays the overlap and distinct OTUs identified in the fecal samples of DP and WB. In this study, OTUs serve as preliminary classification units within the AmpliSeq approach, used for initial grouping before further taxonomic assignment to specific species or genera.

**Figure 2 animals-15-00747-f002:**
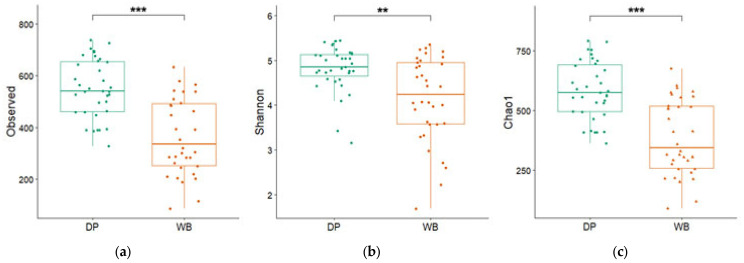
The alpha diversity of pigs’ gut microbiota. (**a**) Observed species index, (**b**) Shannon index, and (**c**) Chao1 index are included. The green and orange bars represent domestic pigs and wild boars, respectively. DP, domestic pig; WB, wild boar. ** *p* < 0.01, *** *p* < 0.001.

**Figure 3 animals-15-00747-f003:**
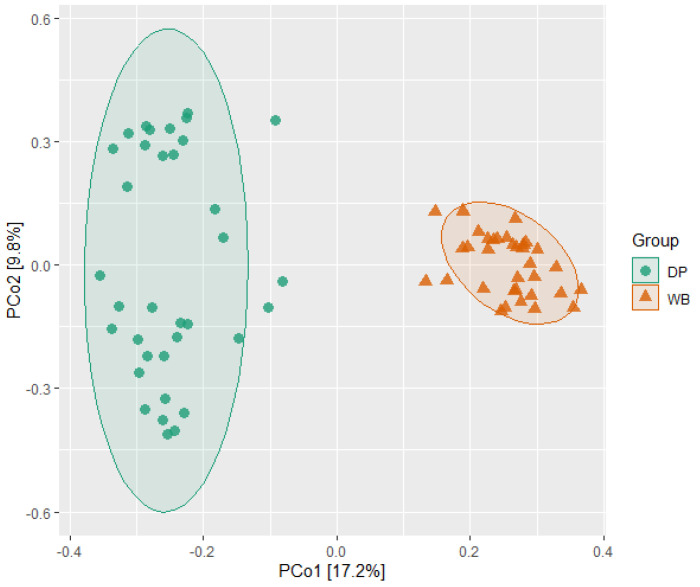
Principal coordinates analysis (PCoA) plot of beta diversity of pigs’ gut microbiota. Each point represents a sample, and two clusters indicated a significant difference in the bacterial community between the two groups. DP, domestic pig; WB, wild boar.

**Figure 4 animals-15-00747-f004:**
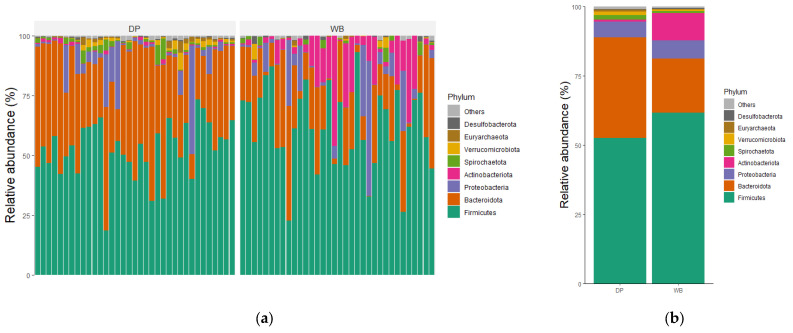
Relative abundance of microbial community in DP and WB groups at (**a**,**b**) phylum, (**c**,**d**) family, and (**e**,**f**) genus levels. (**a**,**c**,**e**) The gut bacterial composition of each subject; (**b**,**d**,**f**) the gut bacterial composition of each group representing the mean relative abundance. DP, domestic pig; WB, wild boar.

**Figure 5 animals-15-00747-f005:**
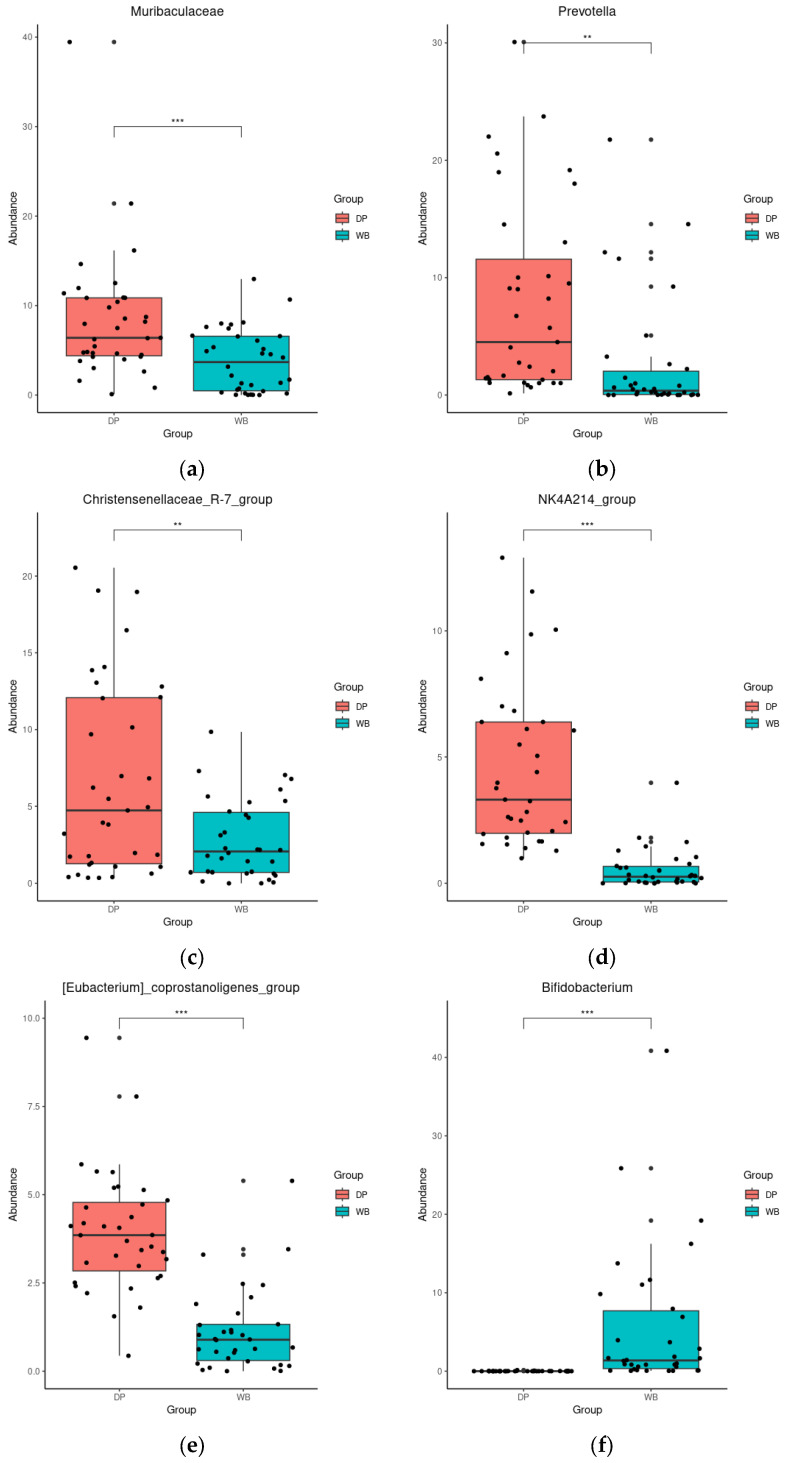
Differential abundance analysis of bacterial genera with DESeq2. (**a**): *Muribaculaceae*, (**b**): *Prevotella*, (**c**): *Christensenellaceae_R-7_group*, (**d**): *NK4A214_group*, (**e**): *[Eubacterium]_coprostanoligenes_group*, (**f**): *Bifidobacterium*, (**g**): *Blautia*, (**h**): *Escherichia-Shigella*, (**i**): *UCG-008*, (**j**): *Lactobacillus*, ** *p* < 0.01, *** *p* < 0.001. The red and blue bars represent domestic pigs and wild boars, respectively. DP, domestic pig; WB, wild boar.

**Figure 6 animals-15-00747-f006:**
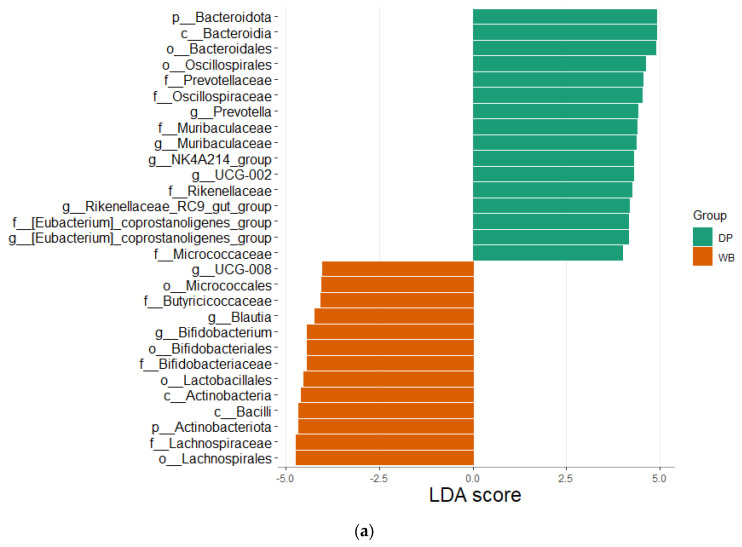
Linear discriminant analysis effect size (LEfSe) analysis. Outputs show the microbial species with significant differences in two groups at the level of phylum, class, order, family, and genus. (**a**) The LEfSe plot shows effect of the size of significantly enriched taxa (LDA score 4). (**b**) The cladogram shows significant taxa. p_, phylum; c_, class; o_, order; f_, family; g_, genus, s_, species. DP, domestic pig; WB, wild boar.

**Figure 7 animals-15-00747-f007:**
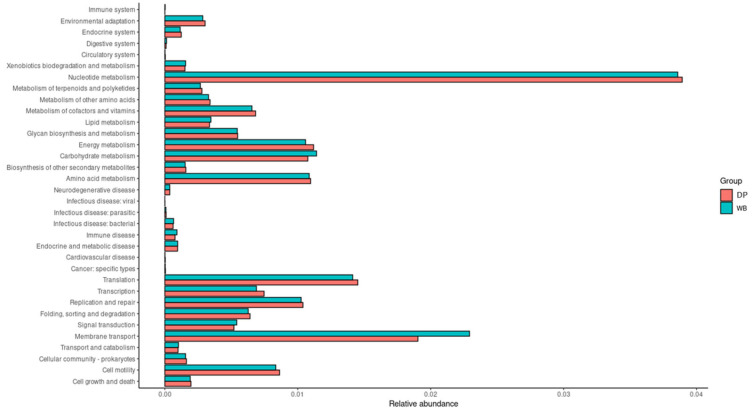
Functional prediction of KEGG pathways at level 2 using PICRUSt2 analysis. In the figure, the abundance of the biological pathways between the two groups is statistically significant (*p* < 0.05). Red and blue represent the DP group and the WB group, respectively. DP, domestic pig; WB, wild boar.

**Table 1 animals-15-00747-t001:** Sample distribution by groups, stage of age, sex, weight, and farm or location of capture.

	Domestic Pig	Total (*n* = 35)	Wild Boar	Total (*n* = 34)
Stage/Age (years)				
	Weanling	8	Juvernile (<1 year)	7
	Grower	8	Subadult (1–2 year)	12
	Gestating sows	8	Adult (>2 year)	15
	Lactating sows	7		
	Finisher	4		
Sex				
Male		15		15
Female		20		19
Farm/Location of capture (Province)				
	Farm A	15	Gyeongnam	15
	Farm B	20	Jeonnam	12
			Gangwon	7

## Data Availability

The original contributions presented in the study are included in the article, further inquiries can be directed to the corresponding author.

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
