# Peer review of "Consequences of Domestication on Gut Microbiome: A Comparative Analysis Between Wild Boars and Domestic Pigs"

_animals, 2025, doi:10.3390/ani15050747_

Round 1

Reviewer 1 Report (Previous Reviewer 1)

Comments and Suggestions for Authors

The manuscript has been significantly improved but some minor issues must be addressed before publication in Animals.

Please refer to the attached file for further comments.

Author Response

Response to reviewers’ comments

We are pleased to resubmit a revised manuscript entitled “Consequences of Domestication on Gut Microbiome: A Comparative Analysis Between Wild Boars and Domestic Pigs” for reconsideration in Animals as an original manuscript. We have carefully evaluated the reviewers’ comments and have provided a point-by-point response below. Changes in the manuscript have been identified by page and sentence, and note by blue FONT. We hope that the revised manuscript meets the reviewers’ expectations at Animals.

Reviewer #1.

The manuscript has been significantly improved but some minor issues must be addressed before publication in Animals.

Please refer to the attached file for further comments: peer-review-44130334.v2.pdf

In the revised file, changes were reflected according to the reviewer’s comment in blue fonts.

line 22, “---- and suggest that wild boar microbiota could be explored for probiotic applications in domestic pigs to improve their health.”

  • This conclusion is not sufficiently supported by the results of the manuscript. I strongly recommend to limit conclusions to what you observe, that is, the differences in composition and the possible causes of these differences.

Answer: the following corrections have been made to the points raised by the reviewer, and reflected in the manuscript.

“In our research, we investigated the distinctions in intestinal microbiota between wild boars and domestic pigs to identify microorganisms absent in the latter. Our study uncovered lactobacilli strains unique to wild boars, which present potential for development as probiotics.”

It should be noted, however, that the efficacy of these strains in enhancing the health of domestic pigs has not yet been empirically validated.

Line 28, Odd sentence. Authors analyse is the gut microbiota via faeces analysis, therefore, I strongly recommend to make it explicit to avoid misunderstanding.

It was revised according to the reviewer’s comment in the revised manuscript. Thank you.

  • This study employed 16S rRNA sequencing to compare the gut microbiota compositions of domestic pigs and wild boars, analyzing 69 fecal samples from both groups.

Line 31-32. Redundant with previous lines.

From line 28~32 commented by the reviewer, those sentences are revised according to the comments as follows: “This study employed 16S rRNA sequencing to compare the gut microbiota compositions of domestic pigs and wild boars, analyzing 69 fecal samples from both groups.”

Line 33, “known for their potential health-promoting effects,” unnecessay in the abstract.

It was revised according to the reviewer’s comment in the revised manuscript. Thank you.

Line 66-68. These lines should be removed.

It was removed according to the reviewer’s comment. Thank you.

line 86. Delete “balanced”

The "balanced" was removed according to the reviewer’s comment. Thank you.

line 105. Rephrase “through a bead-beating method.”

The relevant content has been corrected according to the reviewer’s comment. Thank you.

Line 133. Corrected Illumina iSeq to iSeq 100

It was revised. Thank you.

Line 150-151 [...] population distribution at various taxonomic levels.

The relevant content has been corrected according to the reviewer’s comment. Thank you.

Line 152-153. Alpha diversity indices, namely Shannon-Wiener, Chao1 and Observed Species, were calculated [...]

The relevant content has been corrected according to the reviewer’s comment. Thank you.

Line 193-194 Please make these asterisks bigger.

The relevant figure has been corrected according to the reviewer’s comment. Thank you.

Line 209-210. In this figure, the legend and the axis titles are too small. Please amend accordingly.

The descriptions of the figure have been corrected according to the reviewer’s comment. Thank you.

Line 234-235 Titles, axes labels, and legends are too small to be read properly. Please ammend the figure.

Line 258-259 Idem as in previous figures.

The figures were corrected according to the reviewer’s comment. Thank you.

Line 323 Corrected species to genera.

Revised. Thank you.

Line 331. do authors have this data? If so, it should be included.

Regarding the data on “antibiotic use,” the assertion regarding the impact of antibiotics on the gut microbiota of domestic pigs was not directly investigated in this study. Instead, this assertion is supported by existing literature, as indicated by the citation [41]. We referenced this established research to provide a context for the observed reduction in probiotic diversity in domestic pigs, suggesting that such outcomes could be attributed to factors including simplified diets and conventional breeding practices that typically involve antibiotic use. Thank you.

Line 355, Italics. The relevant terms were revised into Italic. Thank you.  

Line 364-366, This conclusion is not sufficiently supported. Please, do not claim more than you observe that was the significant differences in the microbiota composition between boars and domestic pigs, but nothing related with the introduction of these microorganisms from one species to another.

Please either modify to avoid misunderstanding or delete the sentence.

The conclusion sentence has been revised for clarity and to accurately reflect the observed data without implying unsupported benefits, according to the reviewer’s meticulous comments. Thank you.

Reviewer 2 Report (Previous Reviewer 2)

Comments and Suggestions for Authors

The authors have demonstrably improved the manuscript, incorporating substantial revisions in response to my previous feedback. I acknowledge these improvements with appreciation.

However, one critical point has not been adequately addressed. The finding that Escherichia-Shigella species are more abundant in wild boars, significantly exceeding levels observed in domestic pigs, highlights the inherent risk of exposure to pathogenic microorganisms within the wild environment. The omission of a thorough discussion of this salient issue not only suggests a limited understanding of comparative physiology but also compromises scientific objectivity and rigour.

It is hoped that the authors will acknowledge, assimilate, and critically analyse all relevant findings, at least those of statistical significance, and will conduct thorough discussions to facilitate the generation of novel insights.

Author Response

Response to reviewers’ comments

We are pleased to resubmit a revised manuscript entitled “Consequences of Domestication on Gut Microbiome: A Comparative Analysis Between Wild Boars and Domestic Pigs” for reconsideration in Animals as an original manuscript. We have carefully evaluated the reviewers’ comments and have provided a point-by-point response below. Changes in the manuscript have been identified by page and sentence, and note by blue FONT. We hope that the revised manuscript meets the reviewers’ expectations at Animals.

Reviewer #2.

The authors have demonstrably improved the manuscript, incorporating substantial revisions in response to my previous feedback. I acknowledge these improvements with appreciation. However, one critical point has not been adequately addressed. The finding that Escherichia-Shigella species are more abundant in wild boars, significantly exceeding levels observed in domestic pigs, highlights the inherent risk of exposure to pathogenic microorganisms within the wild environment. The omission of a thorough discussion of this salient issue not only suggests a limited understanding of comparative physiology but also compromises scientific objectivity and rigour.

 It is hoped that the authors will acknowledge, assimilate, and critically analyse all relevant findings, at least those of statistical significance, and will conduct thorough discussions to facilitate the generation of novel insights.

We appreciate the critical observation regarding the higher abundance of Escherichia-Shigella in wild boars compared to domestic pigs, and the potential risk of pathogen exposure this represents. This issue warrants a detailed discussion, which we now include to clarify the implications of our findings.

Firstly, the presence of Escherichia-Shigella in wild boars indicates a unique risk associated with the wild environment (Williams et al., 2023). While Escherichia-Shigella species are known to facilitate digestion and nutrient absorption in wild animals (Zeng et al., 2018), they also contribute to the microbial diversity necessary for dietary fiber breakdown, as observed in studies involving sows where increased fiber intake correlated with higher relative abundance of these bacteria (Li et al., 2023).

Moreover, while Escherichia-Shigella is typically associated with microbial ecological imbalance and gastrointestinal disorders, its prevalence can vary seasonally. This is highlighted by Williams et al. (2023), who document seasonal fluctuations – particularly an increase during summer – in Escherichia-Shigella populations. Chen et al. (2024) further support this by noting that while there is variability in the diversity of Escherichia-Shigella, other factors such as seasonal variations tend to have a more pronounced impact on these populations than previously considered.

Given that our sampling was confined to the spring season, the observed high relative abundance of Escherichia-Shigella may reflect seasonal influences and the inherent pathogenic exposure in the wild environment. This environment, unlike that of domestic pigs which are exposed to fewer pathogens, supports a diverse diet and thus a broader spectrum of microbial interactions.

In conclusion, the significant presence of Escherichia-Shigella in the gut microbiota of wild boars highlights the adaptive responses to their dietary and environmental conditions, underlining the complex interaction between diet, microbial ecology, and pathogen exposure in wild populations. This discussion enhances our understanding of comparative physiology and microbial ecology, affirming the scientific rigor of our study.

References

Chen W, Chen X, Zhang Y, Wu H, Zhao D. Variation on gut microbiota diversity of endangered red pandas (Ailurus fulgens) living in captivity acrosss geographical latitudes. Front Microbiol. 2024 Aug 6;15:1420305.

Li S, Zheng J, He J, Liu H, Huang Y, Huang L, Wang K, Zhao X, Feng B, Che L, Fang Z, Li J, Xu S, Lin Y, Jiang X, Hua L, Zhuo Y, Wu D. Dietary fiber during gestation improves lactational feed intake of sows by modulating gut microbiota. J Anim Sci Biotechnol. 2023 May 5;14(1):65.

Williams, C. E., Brown, A. E., and Williams, C. L. (2023). The role of diet and host species in shaping the seasonal dynamics of the gut microbiome. FEMS Microbiol. Ecol. 99:fiad156.

Zeng Y, Zeng D, Zhou Y, Niu L, Deng J, Li Y, Pu Y, Lin Y, Xu S, Liu Q, Xiong L, Zhou M, Pan K, Jing B, Ni X. Microbial Biogeography Along the Gastrointestinal Tract of a Red Panda. Front Microbiol. 2018 Jul 5;9:1411.

Reviewer 3 Report (New Reviewer)

Comments and Suggestions for Authors

This manuscript used some advanced method to clarify the efects of artificial domestication on the gut microbiota taxa of pigs. This study would provide basis theory for the animal production. Some minor revison is esstential to enhance the quality of scientific paper.

line18 change to 16S rDNA

line30 how many wild boars ? and how many domestic pigs?

line39 please supplement the meaning of this study.

line41 change to next-generation

line100 author should obtain the fecal sample from inside. Because the surface of feces might carried some environmental microbes.

line114 change to "gene"

line126-130 please simply this section

line 139-141 please upload the raw data file to public repository and provide the accession number.

line141-164 please describe the specific tool package in R environment (or other program scipt)

line 166 change to statisitic for sequencing data

line 168 please describe the valid ratio of this sequencing program.

line196 delete repeated  "P<0.001"

line337-345 please make this section concise.

line352-366 please use the same font colour in the full text.

Author Response

Response to reviewers’ comments

We are pleased to resubmit a revised manuscript entitled “Consequences of Domestication on Gut Microbiome: A Comparative Analysis Between Wild Boars and Domestic Pigs” for reconsideration in Animals as an original manuscript. We have carefully evaluated the reviewers’ comments and have provided a point-by-point response below. Changes in the manuscript have been identified by page and sentence, and note by blue FONT. We hope that the revised manuscript meets the reviewers’ expectations at Animals.

Reviewer #3.

This manuscript used some advanced method to clarify the efects of artificial domestication on the gut microbiota taxa of pigs. This study would provide basis theory for the animal production. Some minor revison is esstential to enhance the quality of scientific paper.

line18 change to 16S rDNA

It was revised according to the reviewer’s comment. Thank you.

line30 how many wild boars ? and how many domestic pigs?

A total of 69 fecal samples were collected from 34 wild boars and 35 domestic pigs.

line39 please supplement the meaning of this study.

It was revised according to the reviewer’s comment. Thank you.

  • These findings demonstrate significant differences in the gut microbiota composition between domestic pigs and wild boars, underscoring the impact of domestication on microbial diversity. Further research is necessary to explore the potential of transferring beneficial microbes from wild to domestic populations.

line41 change to next-generation

It was revised according to the reviewer’s comment. Thank you.

line100 author should obtain the fecal sample from inside. Because the surface of feces might carried some environmental microbes.

This context was intended to explain why upper layer feces were taken when sampling in the field, and the inner contents are used when extracting DNA. Thank you.

line114 change to "gene"

It was revised according to the reviewer’s comment. Thank you.

line126-130 please simply this section

It was revised to be simplified according to the reviewer’s comment. Thank you.

line 139-141 please upload the raw data file to public repository and provide the accession number.

The raw data file has been submitted to the NCBI Sequence Read Archive (SRA).

  • SRA submission no.: SUB15114691
  • BioProject: Processed (PRJNA1229100)
  • BioSample: Processing

line141-164 please describe the specific tool package in R environment (or other program scipt)

The package or script used for the analysis has been listed in the subsection 2.4. Thank you.

line 166 change to statisitic for sequencing data

The subheading was revised according to the reviewer’s comment. Thank you.

line 168 please describe the valid ratio of this sequencing program.

All of the statistics for sequencing data including the valid ratio has been described in the subsection 3.1.

line196 delete repeated  "P<0.001"

The repeated one has been deleted. Thank you.

line337-345 please make this section concise.

The relevant sentence has been revised concisely according to the reviewer’s comment. Thank you.

line352-366 please use the same font colour in the full text.

We keep in mind the reviewer’s comment and will reflect the comment when the final version be submitted. Thank you.

This manuscript is a resubmission of an earlier submission. The following is a list of the peer review reports and author responses from that submission.

Round 1

Reviewer 1 Report

Comments and Suggestions for Authors

The manuscript submitted by Bae et al. displays the results of an investigation regarding the comparison of the microbiome between boars and domestic pigs using a 16S rRNA gene metabarcoding approach. The outcomes revealed significant differences in the species’ gut microbiome relative abundances which could be attributed a variety of reasons such as the controlled housing conditions, the diet, etc that could have an impact on their gut health, therefore, the overall wellness of the animal. Such information could be used in animal production to better understand how human intervention actually affects different aspects of animal welfare linked to gut dysbiosis.

Despite the relevance of the study, the manuscript suffers from various major flaws which, unfortunately, make it not to be publishable in its current form. Among them, the most critical is the constant lack on detail throughout all the text along with a poor discussion of the obtained results, which is a pity considering the amount of work undertaken. I strongly recommend the authors to reconsider the whole manuscript, especially the results and discussion sections (which I would advise to merge since it would facilitate the reading) adding more detail. Otherwise, the conclusions drawn are not sufficiently supported remaining as too speculative. Please consider my comments below intended to improve the global quality of the manuscript.

GENERAL COMMENTS

-         Introduction should be better focused on the topic. Reference to previous studies carried out in the field, and an appropriate rationale highlighting the contribution of the present article should be made by the authors.

-         To avoid confusion, scientific names should be used to a subspecies level, i.e., Sus scrofa scrofa and Sus scrofa domesticus for boars and pigs, respectively.

-         According to the current taxonomy guidelines, Firmicutes and Proteobacteria phyla are no longer applicable. Please use Bacillota and Pseudomonadota, respectively. (Int J Syst Evol Microbiol (2021) 71:005056)

SPECIFIC COMMENTS

L25-39 As written the abstract does is not representative of the investigation behind. Please include some data which would render it clearer for the reader.

L35 Without a proper justification, expressions like beneficial microbes are arbitrarily used since there is no description regarding the benefits. Additionally, the term microbe is quite dated and not scientifically appropriate nowadays.

L38-39 Authors did not perform any assay to assess the effects of the boars’ gut microbiota among pigs. Consequently, such statement is merely speculative and must be removed.

L45 Reference needed.

L53-54 Use High-throughput sequencing (HTS) instead of NGS.

L55 These methods… (try to avoid adjectives that could add subjectivity to the text).

L64-65 As written, this statement is not sufficiently justified. Please correct accordingly.

L69 Give specific examples of these beneficial gut bacteria and why authors consider as beneficial.

L100 Total genomic DNA

L112-113 Please correct the protocol using the final concentration used for the PCR reaction.

L113-114 Include the reference for the primers, i.e., Microorganisms (2021) 9:361.

L127 iSeq100

L128 The link for reference 13 does not work.

L121-128 The protocol is incomplete. There is no information about the adapters, the barcode etc. even though on the website provided there is a link with information of the fusion primers (considering that is what authors used), that should be ~89 nucleotides each. Such information must be explicitly displayed in the manuscript, otherwise the reproducibility of the assays is not possible (see: https://help.ezbiocloud.net/wp-content/uploads/2019/02/Chunlab_Bacterial-V4-Fusion-primer-list-3.xlsx, for primer information).

Additionally, no information about the phiX concentration used or the sample concentration loaded into the iSeq100 cartridge. Please include.

L139 using the microeco R package.

L142 Alpha diversity indices…

L142 The OTU is the cluster of similar DNA sequences used for classification. Thus, the level refers to the taxon, i.e., Phylum, Class, Order, etc. Please correct this conceptual error here and throughout the manuscript.

L144-147 Information about the plot of PCoA is not necessary. Please rewrite these lines.

L159-163 As written, the information regarding the mapped sequences and general reaction quality metrics is not clear. Please reconsider the whole fragment and rewrite it for better clarification.

L164 OTUs but at what level?

Figure 1. adds no additional information considering the preceding lines. Additionally, since there is no information regarding the taxonomic level used, it could induce to misunderstanding. Therefore, it should be removed.

Figure 2 Again, the level for the alpha diversity indices should be used. Additionally, even though as a supplementary table, the values for each individual sample should be facilitated by the authors.

L182-185 At what level? Please provide more detail regarding the results obtained. Additionally, authors should undertake some statistic analysis such as a Permutational analysis of variance (PERMANOVA) to compare the data and determine if there is significance between bacteriomes.

L206 It is surprising that Enterobacteriaceae account for such a small percentage of the whole population. This should be further discussed in the following section.

L267-273 These lines are not necessary. Please remove.

L291 greater heterogeneity.

L294 Authors should include more detail about the studies of the mentioned references.

L301-302 The expression beneficial microbes should be either explained or removed.

L305 This part of the discussion refers to the results of boars’ gut microbiome, and reference [34], i.e., Mallo et al. (2010), refer to domestic pigs. Please pay attention to these details otherwise the information given is not accurate.

L313 Do authors have any information about the use of antibiotics among the pig from samples were collected? If so, it should be explicitly described in the materials and methods section and extensively discussed here. If not, please remove these lines.

L313 Why imbalance? Do authors have a model of what a “balanced” microbiome should be to compare with?

L335-341 Considering the previous comments, the conclusions outlined by the authors are not sufficiently supported and, thus, the whole section has to be rewritten.

Author Response

Response to reviewers’ comments

First of all, the authors deeply appreciated reviewers’ time and efforts on the research paper. We are pleased to resubmit a revised manuscript entitled “Consequences of Domestication on Gut Microbiome: A Comparative Analysis Between Wild Boars and Domestic Pigs” for reconsideration in Animals as an original manuscript. We have carefully evaluated the reviewers’ comments and have provided a point-by-point response below. Changes in the manuscript have been identified by blue FONT. We hope that the revised manuscript meets the reviewers’ expectations at Animals.

Reviewer #1

Open Review

Comments and Suggestions for Authors

The manuscript submitted by Bae et al. displays the results of an investigation regarding the comparison of the microbiome between boars and domestic pigs using a 16S rRNA gene metabarcoding approach. The outcomes revealed significant differences in the species’ gut microbiome relative abundances which could be attributed a variety of reasons such as the controlled housing conditions, the diet, etc that could have an impact on their gut health, therefore, the overall wellness of the animal. Such information could be used in animal production to better understand how human intervention actually affects different aspects of animal welfare linked to gut dysbiosis.

Despite the relevance of the study, the manuscript suffers from various major flaws which, unfortunately, make it not to be publishable in its current form. Among them, the most critical is the constant lack on detail throughout all the text along with a poor discussion of the obtained results, which is a pity considering the amount of work undertaken. I strongly recommend the authors to reconsider the whole manuscript, especially the results and discussion sections (which I would advise to merge since it would facilitate the reading) adding more detail. Otherwise, the conclusions drawn are not sufficiently supported remaining as too speculative. Please consider my comments below intended to improve the global quality of the manuscript.

 GENERAL COMMENTS

  • Introduction should be better focused on the topic. Reference to previous studies carried out in the field, and an appropriate rationale highlighting the contribution of the present article should be made by the authors.

Thank you for comments. The introduction has been revised according to the reviewer’s comment.

  • To avoid confusion, scientific names should be used to a subspecies level, i.e., Sus scrofa scrofaand Sus scrofa domesticus for boars and pigs, respectively.

Those scientific names were revised accordingly. Thank you.

  • According to the current taxonomy guidelines, Firmicutes and Proteobacteria phyla are no longer applicable. Please use Bacillota and Pseudomonadota, respectively. (Int J Syst Evol Microbiol (2021) 71:005056)

The taxonomy was revised accordingly. Thank you.

SPECIFIC COMMENTS

L25-39 As written the abstract does is not representative of the investigation behind. Please include some data which would render it clearer for the reader.

The abstract was revised to more represent our results according to the reviewer’s comment. Thank you.

L35 Without a proper justification, expressions like beneficial microbes are arbitrarily used since there is no description regarding the benefits. Additionally, the term microbe is quite dated and not scientifically appropriate nowadays.

Thank you for the reviewer’s meticulous comments. It has been revised according to the reviewer’s comment.

L38-39 Authors did not perform any assay to assess the effects of the boars’ gut microbiota among pigs. Consequently, such statement is merely speculative and must be removed.

It was removed according to the reviewer’s comment. Thank you.

L45 Reference needed. The corresponding reference was added. Thank you.

L53-54 Use High-throughput sequencing (HTS) instead of NGS.

It was revised. Thank you.

L55 These methods… (try to avoid adjectives that could add subjectivity to the text).

It was revised according to the reviewer’s comment. Thank you.

L64-65 As written, this statement is not sufficiently justified. Please correct accordingly.

It was revised according to the reviewer’s comment. Thank you.

L69 Give specific examples of these beneficial gut bacteria and why authors consider as beneficial: It was revised according to the reviewer’s comment. Thank you.

L100 Total genomic DNA: It was revised according to the reviewer’s comment. Thank you.

L112-113 Please correct the protocol using the final concentration used for the PCR reaction.

It has been rephrased according to the reviewer’s comment. Thank you.

L113-114 Include the reference for the primers, i.e., Microorganisms (2021) 9:361.

The reference was added. Thank you.

L127 iSeq100: it was changed.

L128 The link for reference 13 does not work.

I’m afraid to say but it works on all authors’ computer system.

L121-128 The protocol is incomplete. There is no information about the adapters, the barcode etc. even though on the website provided there is a link with information of the fusion primers (considering that is what authors used), that should be ~89 nucleotides each. Such information must be explicitly displayed in the manuscript, otherwise the reproducibility of the assays is not possible (see: https://help.ezbiocloud.net/wp-content/uploads/2019/02/Chunlab_Bacterial-V4-Fusion-primer-list-3.xlsx, for primer information).

Additionally, no information about the phiX concentration used or the sample concentration loaded into the iSeq100 cartridge. Please include.

All information the reviewer commented was revised in the manuscript. Thank you.

L139 using the microeco R package: It was revised according to the reviewer’s comment.

L142 Alpha diversity indices… It was revised. Thank you.

L142 The OTU is the cluster of similar DNA sequences used for classification. Thus, the level refers to the taxon, i.e., Phylum, Class, Order, etc. Please correct this conceptual error here and throughout the manuscript.

The relevant information has been corrected according to the reviewer’s comment.

L144-147 Information about the plot of PCoA is not necessary. Please rewrite these lines.

It was revised according to the reviewer’s comment. Thank you.

L159-163 As written, the information regarding the mapped sequences and general reaction quality metrics is not clear. Please reconsider the whole fragment and rewrite it for better clarification.

It has been rephrased according to the reviewer’s advice. Thank you.

L164 OTUs but at what level?

Figure 1. adds no additional information considering the preceding lines. Additionally, since there is no information regarding the taxonomic level used, it could induce to misunderstanding. Therefore, it should be removed.

Figure 2 Again, the level for the alpha diversity indices should be used. Additionally, even though as a supplementary table, the values for each individual sample should be facilitated by the authors.

L182-185 At what level? Please provide more detail regarding the results obtained. Additionally, authors should undertake some statistic analysis such as a Permutational analysis of variance (PERMANOVA) to compare the data and determine if there is significance between bacteriomes.

The four comments from the reviewer are all saying the same thing, so the answer was done all at once. In this analysis, OTUs are employed as preliminary classification unit within the AmpliSeq approach, functioning as a basis for further taxonomic assignment to species or genus levels. It has also been reflected and modified in the text.

L206 It is surprising that Enterobacteriaceae account for such a small percentage of the whole population. This should be further discussed in the following section.

Thank you for the reviewer’s meticulous comment. After receiving that opinion, we also looked up various literatures and found that there were opposing results, so authors decided to simply describe as the result of our current research. If we leave one of the references we referred to here, it is as follows:

Vedel G et al. Exploring the potential links between gut microbiota composition and natural populations management in wild boar (Sus scrofa). Microbiological Research 274 (2023) 127444.

L267-273 These lines are not necessary. Please remove.

It was removed. Thank you.

L291 greater heterogeneity. It was changed. Thank you.

L294 Authors should include more detail about the studies of the mentioned references.

Additional details were added in the phrase. Thank you.

L301-302 The expression beneficial microbes should be either explained or removed.

Additional content has been added to that part. Thank you.

L305 This part of the discussion refers to the results of boars’ gut microbiome, and reference [34], i.e., Mallo et al. (2010), refer to domestic pigs. Please pay attention to these details otherwise the information given is not accurate.

Since such a probiotic dose experiment cannot be conducted on wild boars, the authors agreed that this paragraph should be left as is, referring to the results of experiments on domesticated pigs, which are the same species as wild boars. Please read it again.

L313 Do authors have any information about the use of antibiotics among the pig from samples were collected? If so, it should be explicitly described in the materials and methods section and extensively discussed here. If not, please remove these lines.

The lines were modified to avoid confusion. Thank you.

L313 Why imbalance? Do authors have a model of what a “balanced” microbiome should be to compare with?

This part is a literature review, and many literatures mention that intestinal microbiome imbalance causes numerous diseases. As mentioned briefly earlier, it is known that simplified diets and breeding styles including antibiotic use cause intestinal microbiome imbalance.

L335-341 Considering the previous comments, the conclusions outlined by the authors are not sufficiently supported and, thus, the whole section has to be rewritten.

The reviewer is definitely correct. The conclusion was rewritten according to the whole section of the manuscript. Thank you.

Reviewer 2 Report

Comments and Suggestions for Authors

This manuscript presents a study comparing the gut microbiome composition in domestic pigs and wild boars. This research topic is of considerable importance for understanding the microbiome alterations that result from dietary and environmental changes during domestication, and it offers potential insights for advancements in zoology, comparative physiology and nutrition, and the swine industry. The study appears promising, with a seemingly acceptable experimental design and data analysis, well-presented graphics, and standardised writing. However, several significant deficiencies have compromised the overall academic quality and value of the manuscript. Notably, many of the concepts and citations are outdated or have been superseded, leading to an inadequate interpretation of the findings and a lack of depth in the discussion of their mechanisms and significance. Many sentences lack of references. While environmental and dietary influences are frequently mentioned, they are neither analysed nor explicitly hypothesised. This, however, is readily addressed, as correlational analyses could assist in establishing connections from a theoretical or statistical standpoint. Specific comments are detailed below.

Abstract

L27: Despite being subtle, there is still a difference between the gut and intestinal microbiota. Please maintain consistency and check throughout the manuscript.

L29-32: Grammar error, please revise.

L35: Please note that reduced diversity may cause ambiguity, it is suggested to use reduced abundance/levels.

Introduction

L44-45: Please add references.

L51-58: This paragraph discusses obsolete concepts, it is recommended to replace them with the latest issues. And please add references for each sentence and/or perspective.

L68-71: Again, references.

M&M

Section 2.1: A major deficiency points to the lack of dietary information, which is one of the most important factors influencing the composition and function of the microbiome.

Results

L175-178: Grammar errors.

Discussion

L267-279: While the content presented here is more specific, it nevertheless constitutes a repetition of material already covered in the introduction. It is therefore suggested that this information be integrated into the introduction section, with only a concise summary provided at this juncture, allowing the discussion to commence directly.

L281-288: The scope of influence extends beyond these points. Apart from nutritional balance and feed additives, variations in macronutrient level are a key factor which can substantially modulate the compositional and functional characteristics of the gut microbiome. And again, references.

L289-294: The underlying reason of the difference in beta diversity is more or less similar to the alpha, it is recommended to combine the two sections.

L300-318: References. It is necessary to acknowledge that the effects and mechanisms of probiotic supplementation and those of colonised microbiota can differ markedly. Whilst some of the bacteria mentioned are widely used as probiotics, they are also commonly encountered as colonisers, and their fundamental role in the gut remains carbohydrate metabolism and the fermentation of dietary fibre. As noted above, it is likely that differences in macronutrient content underlie these changes. The authors have also mentioned that wild boars potentially have a higher fibre or carbohydrate intake, which leads to a greater production of SCFAs and/or other metabolites. However, this was not been studied or discussed in depth. As mentioned in the general comments, this is better to be indicated by a correlation analysis between diet and differential bacteria, and is not difficult.

On the contrary, the increased abundance of Escherichia-Shigella, a genus typically associated with pathogenic bacteria or opportunistic pathogens, has been completely disregarded. This observation is crucial, since it serves to reflect the susceptibility of wild populations to pathogen exposure and infectious diseases.

L322-325: References. And in-depth discussion is needed, given the bigger picture of the functional changes in the altered microbiome.

Author Response

Response to reviewers’ comments

First of all, the authors deeply appreciated reviewers’ time and efforts on the research paper. We are pleased to resubmit a revised manuscript entitled “Consequences of Domestication on Gut Microbiome: A Comparative Analysis Between Wild Boars and Domestic Pigs” for reconsideration in Animals as an original manuscript. We have carefully evaluated the reviewers’ comments and have provided a point-by-point response below. Changes in the manuscript have been identified by blue FONT. We hope that the revised manuscript meets the reviewers’ expectations at Animals.

Reviewer #2.

Open Review

Comments and Suggestions for Authors

This manuscript presents a study comparing the gut microbiome composition in domestic pigs and wild boars. This research topic is of considerable importance for understanding the microbiome alterations that result from dietary and environmental changes during domestication, and it offers potential insights for advancements in zoology, comparative physiology and nutrition, and the swine industry. The study appears promising, with a seemingly acceptable experimental design and data analysis, well-presented graphics, and standardised writing. However, several significant deficiencies have compromised the overall academic quality and value of the manuscript. Notably, many of the concepts and citations are outdated or have been superseded, leading to an inadequate interpretation of the findings and a lack of depth in the discussion of their mechanisms and significance. Many sentences lack of references. While environmental and dietary influences are frequently mentioned, they are neither analysed nor explicitly hypothesised. This, however, is readily addressed, as correlational analyses could assist in establishing connections from a theoretical or statistical standpoint. Specific comments are detailed below.

Abstract

L27: Despite being subtle, there is still a difference between the ‘gut’ and ‘intestinal’ microbiota. Please maintain consistency and check throughout the manuscript.

After authors discussed over this, gut microbiota was decided to be used and revised in the manuscript. Thank you.

L29-32: Grammar error, please revise. It was revised. Thank you.

L35: Please note that “reduced diversity” may cause ambiguity, it is suggested to use “reduced abundance/levels”.

Thank you for the comment. The other reviewer also said something similar, so some changes were made.

Introduction

L44-45: Please add references.

A reference was added according to the reviewer’s comment. Thank you.

L51-58: This paragraph discusses obsolete concepts, it is recommended to replace them with the latest issues. And please add references for each sentence and/or perspective.

The reviewer could think in that way, but we’d afraid that it would be find as it is now, because we started with “traditionally”. If the reviewer doesn’t mind, please check one more time and let us know if you still need to make corrections.

L68-71: Again, references.

A reference was added according to the reviewer’s comment. Thank you.

M&M

Section 2.1: A major deficiency points to the lack of dietary information, which is one of the most important factors influencing the composition and function of the microbiome.

We have supplemented the information on the feed of the pigs raised.

Results

L175-178: Grammar errors. It was revised. Thank you.

Discussion

L267-279: While the content presented here is more specific, it nevertheless constitutes a repetition of material already covered in the introduction. It is therefore suggested that this information be integrated into the introduction section, with only a concise summary provided at this juncture, allowing the discussion to commence directly.

Those have been rephrased according to the reviewer’s comment. Thank you.

L281-288: The scope of influence extends beyond these points. Apart from nutritional balance and feed additives, variations in macronutrient level are a key factor which can substantially modulate the compositional and functional characteristics of the gut microbiome. And again, references.

Thank you for your constructive comments. We appreciate your insights regarding the influence of macronutrient levels on the gut microbiome of domestic pigs. The main reason why domesticated pigs have a high alpha-diversity is because of the nutrient-balanced diets and farming practices, which are described and referenced accordingly.

L289-294: The underlying reason of the difference in beta diversity is more or less similar to the alpha, it is recommended to combine the two sections.

We respect the reviewer's opinion very much, but there were some additions made during our revision process, so we couldn't combine the sections. We ask that you please read it again.

L300-318: References. It is necessary to acknowledge that the effects and mechanisms of probiotic supplementation and those of colonised microbiota can differ markedly. Whilst some of the bacteria mentioned are widely used as probiotics, they are also commonly encountered as colonisers, and their fundamental role in the gut remains carbohydrate metabolism and the fermentation of dietary fibre. As noted above, it is likely that differences in macronutrient content underlie these changes. The authors have also mentioned that wild boars potentially have a higher fibre or carbohydrate intake, which leads to a greater production of SCFAs and/or other metabolites. However, this was not been studied or discussed in depth. As mentioned in the general comments, this is better to be indicated by a correlation analysis between diet and differential bacteria, and is not difficult.

On the contrary, the increased abundance of Escherichia-Shigella, a genus typically associated with pathogenic bacteria or opportunistic pathogens, has been completely disregarded. This observation is crucial, since it serves to reflect the susceptibility of wild populations to pathogen exposure and infectious diseases.

While we prepared resubmission, some degree of revision was conducted. If the author kindly read again and find some more details to be revised, please let us know. I appreciated deeply the reviewer’s time and effort on this manuscript.

L322-325: References. And in-depth discussion is needed, given the bigger picture of the functional changes in the altered microbiome.

The discussion and conclusion sections were rephrased according to the reviewer’s constructive advice. Thank you.

Reviewer 3 Report

Comments and Suggestions for Authors

Dear authors:

The presented paper is a very interesting document about the comparative gut microbiota profile of domestic and wild pigs, that conducted you to the obtention of very useful results which could direct your next future works to the isolation of strains with probiotic potentials.

In my opinion your work has a good study design, an adequate methodology and highly satisfactory results.

Author Response

Response to reviewers’ comments

First of all, the authors deeply appreciated reviewers’ time and efforts on the research paper. We are pleased to resubmit a revised manuscript entitled “Consequences of Domestication on Gut Microbiome: A Comparative Analysis Between Wild Boars and Domestic Pigs” for reconsideration in Animals as an original manuscript. We have carefully evaluated the reviewers’ comments and have provided a point-by-point response below. Changes in the manuscript have been identified by blue FONT. We hope that the revised manuscript meets the reviewers’ expectations at Animals.

Reviewer 3

Open Review

Comments and Suggestions for Authors

Dear authors:

The presented paper is a very interesting document about the comparative gut microbiota profile of domestic and wild pigs, that conducted you to the obtention of very useful results which could direct your next future works to the isolation of strains with probiotic potentials.

In my opinion your work has a good study design, an adequate methodology and highly satisfactory results.

Thank you.